# Triage Grading and Correct Diagnosis Are Critical for the Emergency Treatment of Anaphylaxis

**DOI:** 10.3390/children9121794

**Published:** 2022-11-23

**Authors:** Arianna Dondi, Elisabetta Calamelli, Sara Scarpini, Egidio Candela, Giovanni Battista Biserni, Chiara Ghizzi, Francesca Lombardi, Paola Salvago, Laura Serra, Ilaria Corsini, Marcello Lanari

**Affiliations:** 1Pediatric Emergency Unit, IRCCS Azienda Ospedaliero-Universitaria di Bologna, 40138 Bologna, Bologna, Italy; 2Pediatric and Neonatology Unit, Imola Hospital, 40026 Imola, Bologna, Italy; 3Specialty School of Pediatrics, Alma Mater Studiorum, University of Bologna, 40138 Bologna, Bologna, Italy; 4Pediatric Unit, Maggiore Hospital, 40133 Bologna, Bologna, Italy

**Keywords:** anaphylaxis, emergency service, hypersensitivity, epinephrine

## Abstract

Introduction: Anaphylaxis is one of the most frequent and misdiagnosed emergencies in the pediatric emergency department (PED). We aimed to assess which factors play a major role for a correct diagnosis and an appropriate therapy. Methods: We reviewed the records of children discharged with a diagnosis of anaphylaxis or an allergic reaction over 11 years from 3 hospitals in the Bologna city area. Results: One hundred and sixteen cases matched the criteria (0.03% of the total admittances) and were divided according to the patients’ symptoms at arrival: active acute patients [AP], *n* = 50, or non-acute patients ([NAP], *n* = 66). At the patients’ discharge, anaphylaxis was diagnosed in 39 patients (33.6%). Some features seemed to favor a correct diagnosis: active symptoms at arrival (AP vs. NAP, *p* < 0.01), high-priority triage code (*p* < 0.01), and upper airway involvement (*p* < 0.01). Only 14 patients (12.1%), all in the AP group, received epinephrine, that was more likely administered to patients recognized to have anaphylaxis (*p* < 0.01) and with cardiovascular, respiratory, or persistent gastrointestinal symptoms (*p* < 0.02), as confirmed by logistic regression analysis. Conclusions: Anaphylaxis is still under-recognized and under-treated. Correct triage coding and a proper diagnosis seem to foster an appropriate treatment. Physicians often prefer third-line interventions. Specific training for nurses and physicians might improve the management of this disease.

## 1. Introduction

Anaphylaxis is a severe, potentially life-threatening systemic hypersensitivity reaction characterized by rapidly developing airway, breathing, or circulatory problems usually associated with skin or mucosal changes [1,2]. Its incidence in Europe is around 1.5–7.9 per 100,000 person–years and it has an overall fatality rate below 0.001% [3]. In the pediatric emergency department (PED), it is one of the most frequent emergencies with a rate of presentations up to 1:1000 [4].

Anaphylaxis often undergoes a spontaneous resolution thanks to endogenous mechanisms, but its course and progression are unpredictable, and even mild symptoms can quickly evolve into cardio-respiratory arrest [5,6]. 

The most important drug in the treatment of anaphylaxis is intramuscular (IM) epinephrine [1,2,7]. It has an excellent safety profile: mild side effects (transient pallor, palpitations, and headache) can happen due to its mechanism of action, but the severe ones (arrhythmias, myocardial infarction, pulmonary edema, and intracranial hemorrhage) are extremely rare and usually due to overdosing, an inappropriate use, or an intravenous administration [1]. Conversely, antihistamines and corticosteroids, commonly used in EDs, are third-line interventions, the former being useful to relieve cutaneous symptoms and the latter possibly preventing protracted symptoms mainly in asthmatic patients [1,2].

The overall prognosis of anaphylaxis is good, but still, in the UK, deaths due to anaphylaxis are approximately 20 each year [2]. A failed, delayed, or inappropriate administration of epinephrine is the main cause of a fatal outcome; an immediate recognition and a prompt treatment are thus crucial [8,9].

Medical evidence proves both the practice and knowledge gaps about anaphylaxis for ED professionals. Apparently, up to 50% of cases are misdiagnosed in the ED [10] and IM epinephrine is underused in this setting [11].

The present study was performed to assess which factors play a major role in performing a correct diagnosis and administering the appropriate therapy for anaphylaxis in the PED.

## 2. Materials and Methods

### 2.1. Study Design and Patients

This is a retrospective, multicenter study of children presenting with anaphylaxis over an 11-year period to the PED of 3 hospitals in the Bologna city area (Northern Italy).

In all the centers, patients were accepted by a triage-qualified nurse who defines acuity by a 4-grade color scale (white, green, yellow, and red; Table 1); yellow and red tags are high-priority codes. 

For each patient, a record form is filled in by the nurse and medical staff, including the demographic data, all the relevant clinical information, performed tests, administered therapy, clinical re-evaluation, diagnosis, discharge modality, and home therapy prescription; the same form includes data about a short-stay observation, if done. 

We retrospectively reviewed the records of children discharged with a diagnosis of anaphylaxis or an allergic reaction from January 2008 to December 2018 for S. Orsola-Malpighi Hospital, and from January 2011 to December 2018 for Maggiore and Imola Hospitals because older files could not be retrieved by the informatics system. Cases were identified from the medical record databases using the search terms “anaph*” or “allerg*” and screened only if the reported information was sufficient to confirm or exclude a diagnosis of anaphylaxis; those matching the EAACI criteria for anaphylaxis [1] were considered eligible for inclusion. The medical records were independently analyzed by at least two physicians (one trainee in pediatrics, SS, and a pediatrician with expertise in pediatric emergency medicine and pediatric allergy, ElC); in case of a disagreement, cases were discussed until a common decision was reached. To reduce the risk of errors in the data extraction, this process was performed by one of the authors (EgC) and its accuracy was then checked by another author (AD), who verified all the charts.

### 2.2. Statistical Analysis

The sample size was defined by the number of eligible charts. A descriptive analysis was performed for continuous variables (mean and standard deviation, SD). The associations between the qualitative variables were evaluated with a Chi-square test or Fisher exact test. Student’s t test was used to compare the means. A binary logistic regression was performed to outline the factors mostly associated with an IM epinephrine administration, which was the dependent variable; the symptoms at the onset, the categories of the allergens, a previous history of anaphylaxis, and the progression of the symptoms were the independent variables. A similar analysis was performed to explore the factors mostly associated with the diagnosis of anaphylaxis; the symptoms at the onset, the categories of the allergens, the symptoms’ progression, and the priority code in the PED were the independent variables. The same regressions were also undertaken to investigate whether the association of the concurrent symptoms (two at a time, for instance gastrointestinal and cutaneous, or cardiovascular and respiratory symptoms) and the aforementioned independent variables could be associated with an epinephrine administration and diagnosis of anaphylaxis. The results were deemed as significant for *p* < 0.05. STATA 7.0 (Stata Corporation, College Station, TX, USA) was used for the analysis.

### 2.3. Ethics

This study is part of a larger protocol approved by the S. Orsola Hospital Ethical Reviewer Board (number 8/2019/Oss/AOUBo). The research was conducted in accordance with the Helsinki Declaration.

## 3. Results

During the years of the study, the three PEDs registered 429.652 visits. Among them, 116 patients (0.03% of the total) met the EAACI criteria for anaphylaxis (Figure 1). 

The main features of the study population are described in Table 2. 

The mean age of the patients was 6.6 years (range: 0.4–21.3 years); 24% were younger than 2 (Figure 2). 

The patients were divided into two groups: (1) children with clear ongoing anaphylaxis symptoms when arriving in the PED (acute patients, [AP], *n* = 50, 43.1%), and (2) children whose symptoms were in a recovery phase at arrival (non-acute patients, [NAP], *n* = 66, 56.9%). The mean age of the NAP was 5.5 years, while that of the AP was 8.0 years (*p* < 0.01). In the AP group, most of the children were triaged as high priority, but 26% were given a green or white tag; among the NAP, only 24% received a yellow code and none a red one. Thirty-three patients (28.5%) arrived at the PED by ambulance, with no s.s. differences between the AP and NAP.

### 3.1. Suspected Triggers

The suspected triggers of the anaphylactic reactions are described in Figure 3.

Most of the episodes were likely triggered by foods (*n* = 72, 62%), mainly cow’s milk and nuts, followed by drugs in 17 (15%) reactions. In 19% of the events (*n* = 22), it was not possible to state the etiological agents by the patients’ medical history.

### 3.2. Symptoms and Clinical Manifestations

The skin was the most affected organ (*n* = 110, 94.8%), followed by the lower respiratory tract (*n* = 74, 63.8%); almost half of the patients presented with gastrointestinal (*n* = 55, 47.4%) or upper airway compromise (*n* = 51, 44%), whereas the cardiovascular system was involved only in a small percentage (*n* = 7, 6%) (Table 3).

The presence of cardiovascular symptoms made an IM epinephrine administration more likely: 4 out of 5 patients with these symptoms in the AP group received the drug vs. 10/45 who did not have a cardiovascular involvement (*p* < 0.01). Cardiovascular and respiratory symptoms occurred mainly in the AP group, while gastrointestinal symptoms were predominant among the NAP (n.s.).

### 3.3. Diagnosis of Anaphylaxis

Seventy-seven children received a diagnosis of an allergic reaction (66.4%), while only 39 patients were recognized to have an anaphylactic reaction (33.6%), as reported in Table 4. 

We identified some features that seemed to increase the probability of a correct diagnosis: the “acute” presentation with ongoing anaphylactic symptoms (32/50, 64%, in the AP vs. 7/66, 10.6%, in the NAP group, *p* < 0.01), the assignment of a yellow or red tag at triage (27/53 children with a high-priority code vs. 12/43 with a white/green code, *p* < 0.001), and the presence of active signs of upper airway involvement (13/21 patients with active upper airway involvement vs. 26/95 without such symptoms, *p* < 0.01). The diagnosis was also apparently easier among the 20 patients with previous anaphylaxis episodes, however, 7 of them (35%) were diagnosed with an “allergic reaction”.

### 3.4. Treatment of Anaphylaxis

Only 14 patients (12.1%) were treated with IM epinephrine, and they were all part of the AP group. Corticosteroids and antihistamines were used in 84% of the patients in the AP group and in almost half of the NAP. Overall, 74 patients (63.8%) were only treated with an antihistamine and/or a corticosteroid, with no significant differences between the two groups. None of the patients had received IM epinephrine before arrival to the PED, whereas 52 (44.8%) had taken antihistamines and/or corticosteroids. Twenty-eight (24.1%) patients received no treatment at all in the PED and this was more evident among the NAP group (Table 4); most of them, however, had been administered an antihistamine or a corticosteroid before their arrival. 

Among the patients who received the diagnosis of anaphylaxis, the administration of IM epinephrine (12/39 patients, 30.8%) was more likely than in those who did not receive the correct diagnosis (2/77, 2.6%, *p* < 0.01). Despite the administration of the gold-standard, specific therapy, two patients who were treated with IM epinephrine were discharged with a diagnosis of an “allergic reaction” instead of anaphylaxis. Only 4/20 children with a known anaphylaxis history were treated with IM epinephrine. 

No patients receiving IM epinephrine experienced any severe side effect.

### 3.5. Discharge from the PED

The patients in the AP group were more likely to be kept in a short-stay observation rather than those in the NAP group (*p* < 0.01), who were more frequently discharged within 6 h from arrival to the PED (Table 4). Only three patients required admission.

An allergological evaluation was performed or planned in 47 patients (40.5%) and recommended to 17 (14.7%). Forty-nine children (42.2%) received no written instructions about this. The opinion of the allergist was requested more frequently for the AP (42/50, 84%) than for the NAP (25/66, 37.9%) (*p* < 0.01). An epinephrine auto-injector (EAI) was recommended at the PED discharge only to 10.8% (*n* = 11) of the children who did not already have one. Most patients were discharged with a prescription of antihistamines or corticosteroids.

### 3.6. Logistic Regression

Logistic regression analysis confirmed a positive correlation between the presence of ongoing cardiovascular, respiratory, and persistent gastrointestinal symptoms, but not cutaneous symptoms and the administration of IM epinephrine. Conversely, there was no association between the symptoms in a recovery phase at the patients’ arrival and a treatment with IM epinephrine. Moreover, the type of suspected allergen and the occurrence of previous anaphylactic reactions did not correlate with the use of epinephrine. This model significantly explains 65% of IM adrenaline epinephrine administrations in our patients (Table 5).

The second logistic regression predicts the diagnosis of anaphylaxis in 37.5% of cases and the strongest associations were with the priority code given at the PED admission and the persistence and/or progression of symptoms. Regression models with the presence of two multiple concurrent symptoms are not better predictors of neither an epinephrine administration nor a diagnosis of anaphylaxis, more than the single symptoms themselves.

## 4. Discussion

The present paper analyses 116 visits for anaphylactic reactions in 3 PEDs, representing 0.03% of all the PED visits. This data are lower than what reported in the literature: an incidence of 0.2% and 0.1% were found in an American and in a Spanish PED, respectively [12,13].

Two distinct groups of patients could be recognized in our cohort, depending on their clear ongoing symptoms (AP) or with a progressive spontaneous remission (NAP) at the patients’ arrival. These two groups were considered separately to better identify those cases with an acute presentation who were under-recognized, and those who were under-treated despite a correct diagnosis.

Our study shows that less than half of the patients presenting with symptoms evocative of anaphylaxis received a high-priority code, similarly to what happened in a Spanish PED where, among 137 children referred for anaphylaxis, only 56 (33%) were triaged correctly [14]. However, the Canadian Pediatric Emergency Triage Acuity Scale [15] suggests that anaphylaxis should be prioritized to level I or II (resuscitation or emergency) and highlights how current triage, which is based on severity perception, has the limit to under-evaluate most of these cases, thus delaying treatment [14]. Another paper by the same Spanish group investigated the impact of a specific educational intervention (a training lecture to PED triage nurses and design of a reference card highlighting symptoms and risk factors for anaphylaxis). The intervention appeared to be effective in increasing the recognition of anaphylaxis patients and decreasing waiting times [16].

Even if based only on the medical history collected in the PED, our results show that food is the major elicitor of anaphylaxis in children, as described in the European Anaphylaxis Registry [17], and they confirm the increasing trend of tree nuts- and peanut-associated anaphylaxis in Italy [18]. Although the mechanisms underlying this trend are not well understood, the changing habits in food consumption (e.g., vegetarian/vegan diets, use of pre-prepared foods, etc.) may play a role in increasing the exposure to nuts early in childhood [19]. In 20% of the cases, we could not define the presumed elicitors by the patients’ medical history, in line with previous epidemiological studies based only on the ED assessment, where in almost ¼ of cases the identification of the specific trigger was not possible; a further allergological evaluation identifies the eliciting factors in up to 99% of the episodes [20]. 

Similarly to what was reported in the pediatric literature [19], our data show that skin involvement occurs in most cases (94.8%), respiratory and gastrointestinal symptoms in about half, and a cardiovascular involvement only in a small percentage of children, contrarily to what happens in adults [4]. If evaluated according to the clinical presentation, it seemed that the most severe reactions occurred in the older patients: the mean age was significantly higher in the AP group than in the NAP, confirming that older children and adolescents are at a higher risk of severe reactions [1].

Our results confirm that anaphylaxis is still under-treated in the PED setting. Despite the fact that IM epinephrine is the first-line treatment and its therapeutic efficacy and safety has been proved in the pediatric population [1,11,21], in our study, only 12% of the patients received this drug, and even in the AP group, whose patients were presented to the PED with clear ongoing symptoms, less than 30% were properly treated. In agreement with the literature, none of the patients treated with IM epinephrine had any severe side effects. Moreover, in line with previous surveys [22], the association of steroids and antihistamines was the most common therapeutic approach, despite these drugs having been classified as third-line interventions by the international guidelines [1]. In fact, the onset of an action of corticosteroids is slow and their supposed role in preventing biphasic reactions has not been demonstrated [23], and antihistamines can only relieve cutaneous symptoms [1].

Furthermore, we observed that the administration of epinephrine was more likely in patients arriving to the PED with ongoing clinical symptoms and in those with cardiovascular, respiratory, and/or persistent gastrointestinal involvement; basically the most serious events. The regression model found that the decision was made primarily on the ongoing nature of symptoms, except from the cutaneous ones, rather than the previous medical history of the patient or the type of allergen. Given the fact that the recognition of anaphylaxis is the key factor in administering the correct treatment, from the regression model (Table 6), we confirmed that a strong association with the priority code given at the admission and the presence of persistent or progressive symptoms. 

In other words, the models showed that the diagnosis of anaphylaxis was more likely (up to 37.5%) the more the conditions appeared to be severe in the PED. Interestingly, no single nor combined symptoms seem to significantly increase the probability of reaching the diagnosis, and little role is played by the type of allergen the patients were exposed to. This observation highlights that a specific training may be needed for clinicians and healthcare assistants in order to recognize milder presentations of anaphylaxis. This can lead not only to the administration of the correct treatment in the PED, but also to an allergologist’s referral and eventually to the need for auto-injectable epinephrine.

A recent paper by Dubus et al. about children with anaphylaxis attending the ED reports that epinephrine was more often given in the most severe cases [24]; however, the authors argue that this intuitive adaptation was made by pediatricians, who apparently prefer to use the correct drug only in the most severe situations, which contributes to its underuse in pediatric anaphylaxis.

In our study, no patients died. Even if mortality from anaphylaxis is fortunately extremely low [3], deaths from these reactions are due to a missed or delayed administration of IM epinephrine [6,8] and could therefore be easily avoided.

As confirmed by our results, anaphylaxis is under-diagnosed: the diagnosis was correct only in one third of the patients, and in less than 2/3 of the AP group alone. Some years ago, in an online anonymous survey, Russell et al. evaluated how US EDs managed anaphylaxis and found that the primary gaps were a low (or no) utilization of the standard diagnostic criteria and the inconsistent use of epinephrine [10].

A prompt recognition of anaphylaxis affects the administration of the gold-standard therapy [12]. Our data suggest that an arrival to the PED with acute, ongoing symptoms, a high-priority code at triage, and the presence of objective signs of a respiratory involvement can ease the recognition of anaphylaxis in EDs. A paper by Thomson et al. identified similar factors which help in the diagnosis of anaphylaxis, namely a more severe clinical presentation and a higher acuity triage scale; they also found a higher frequency of a correct diagnosis in those arriving by ambulance [25], but we could not confirm this association. 

According to our data, a high percentage of the patients apparently did not receive instructions about an allergy work-up, nor the prescription of an EAI. However, some authors report a recurrence rate of anaphylaxis of almost 20% per year [26]. In fact, a proper management of these patients includes a prescription of an EAI, instructions about the treatment of new episodes, and a thorough evaluation of the triggering allergen with an etiologic treatment, if feasible [27]. 

However, our paper alarmingly revealed that none of the patients who had been previously prescribed an EAI used it before coming to the PED. The patients’ and caregivers’ reluctance in administering epinephrine via the EAI is well-known in the literature. A study by Noimark et al. evaluated 245 anaphylactic reactions among 969 patients recruited in 14 pediatric allergy clinics in the UK and found that an EAI was used only in 16.7% of these reactions [28,29]. Reasons for not using the EAI as prescribe include uncertainty about the actual severity of the anaphylaxis episode and the fear of using epinephrine [28,29,30]. These results highlight the need for the continuous education of teenagers and caregivers regarding pediatric patients with previous anaphylactic reactions, with specific emphasis on the recognition of symptoms and relief of the fear of an epinephrine administration.

As shown by Prince et al., many physician barriers surround the proper use of IM epinephrine: the complexity of a diagnosis of anaphylaxis, which cannot be immediately supported by laboratory tests, the lack of knowledge of an epinephrine administration, and the fear of the potential risks concerning its safety profile [31]. Arroabarren and colleagues found that applying an anaphylaxis protocol substantially improved the management of this emergency in the PED [30] and Barni et al. recently demonstrated the usefulness of high-fidelity simulations in increasing the use of epinephrine for anaphylaxis in the PED [32].

## 5. Limitations

Despite the fact that our discharge records were carefully analyzed by at least two different physicians who applied the same inclusion criteria to ensure a uniform interpretation and that we considered three different centers to make the results more generalizable, our study has several limitations: (1) its retrospective design only allows the analysis of what is reported on the PED discharge sheet, so that some relevant data might be missing; (2) it does not include any following allergological work-up, so that the allergologist’s confirmation or exclusion of the diagnosis of anaphylaxis is not considered, as well as the actual triggers of the reaction; (3) despite being multicenter, the overall number of patients is relatively low; and (4) we did not include the diagnosis of “urticaria” and “angioedema” alone in the selection of the patients’ record, so we might have missed some cases.

## 6. Conclusions

The PED physicians hardly perform the diagnosis of anaphylaxis and apparently prefer the generic definition of an “allergic reaction”. Moreover, they often avoid the use of IM epinephrine and choose third-line interventions. However, a correct recognition of the acuity at triage helps the clinician to perform a proper diagnosis and administer the appropriate treatment. To avoid the risk of unnecessary, even if unlikely, child deaths, the recognition and treatment of anaphylactic reactions in the PED might be improved with specific training for nurses and physicians, including a participation in seminars, congresses, and high-fidelity simulation sessions, and by focusing on triage interventions such as the provision of questionnaires or specific reference cards.

## Figures and Tables

**Figure 1 children-09-01794-f001:**
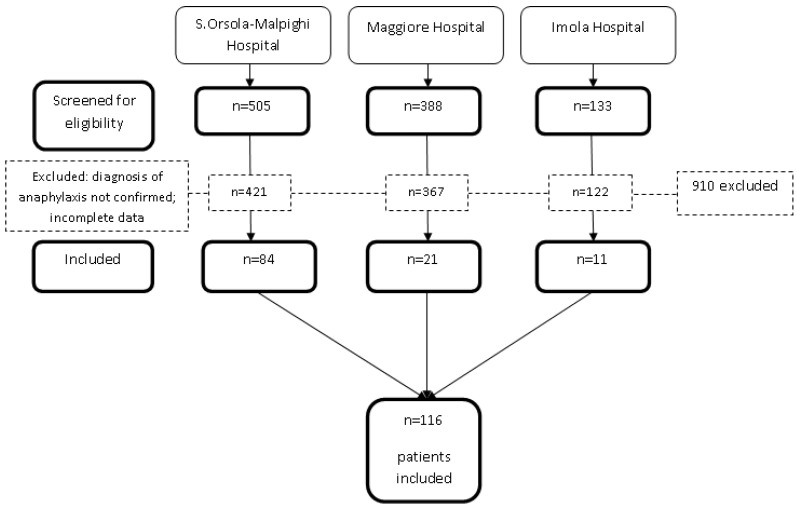
Flow diagram of the study patients.

**Figure 2 children-09-01794-f002:**
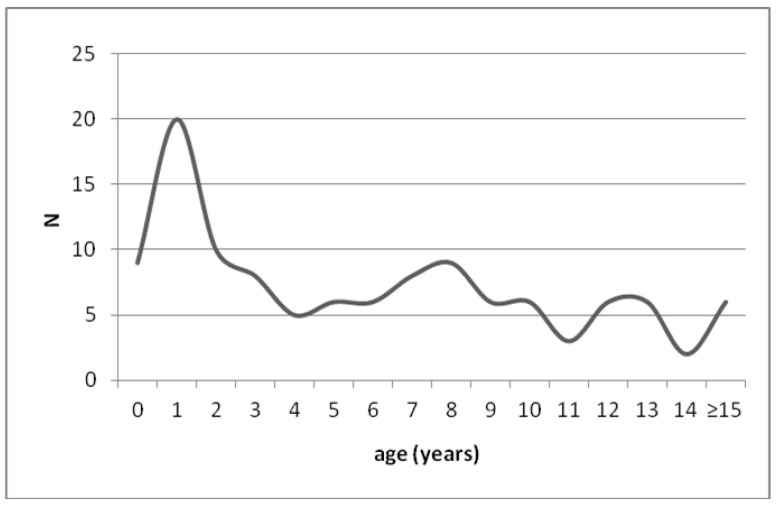
Number of episodes per year of age at PED admission for anaphylaxis.

**Figure 3 children-09-01794-f003:**
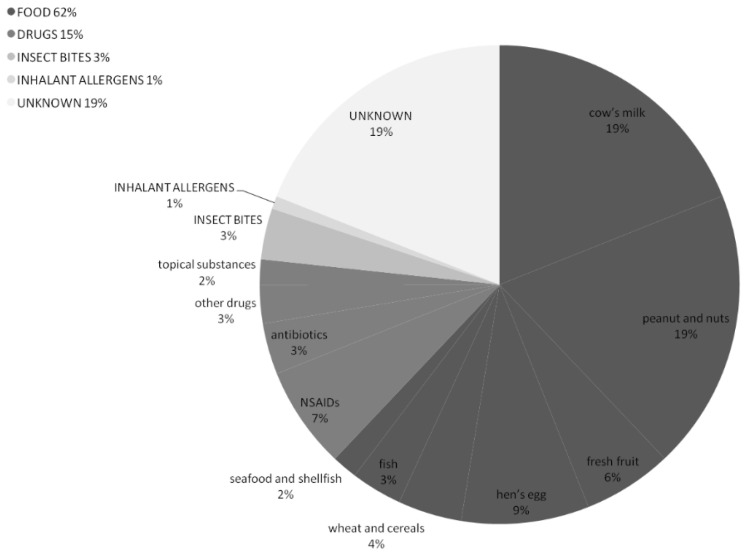
Suspected triggers of the anaphylactic reactions among 116 patients referred to the PED.

**Table 1 children-09-01794-t001:** Triage color code system (Montefiori M, Di Bella E, Leporatti L, Petralia P. Robustness and Effectiveness of the Triage System in the Pediatric Context. Appl Health Econ Health Policy 2017; 15: 795–803).

Color Tag	Features	Priority	Time for Evaluation
White	No acute onset, does not affect vital signs	Non urgent, always inappropriate	Re-assessed by triage nurse if symptoms modify until medical evaluation
Green	Relevant, acute-onset symptoms, normal vital signs	Non urgent, often inappropriate	Re-assessed by triage nurse every 30–60 min until medical evaluation
Yellow	Severe, acute-onset lesions, altered vital signs	Urgent	Closely communicated to the attending physician, rapid medical evaluation
Red	Life-threatening, compromised vital functions	Highly urgent	Immediate medical evaluation

**Table 2 children-09-01794-t002:** Demographic and clinical characteristics of 116 children presenting to the PED for anaphylaxis. AP = acute patients, NAP = non-acute patients.

	Total	AP Group	NAP Group
	*n* = 116	*n* = 50	*n* = 66
**Male, *n* (%)**	76 (65.5)	39 (78)	37 (56.1)
**Age mean (DS)**	6.6 (4.8)	8.0 (5.0)	5.5 (4.5)
**Allergens, *n* (%)**			
Food	72 (62.1)	32 (64)	40 (60.6)
Drugs	15 (12.9)	4 (8)	11 (16.7)
Others	7 (6)	3 (6)	4 (6.1)
Unknown	22 (19)	11 (22)	11 (16.7)
**Previous anaphylactic reaction, *n* (%)**			
Yes	20 (17.2)	14 (28)	6 (9.1)
No	22 (19)	10 (20)	12 (18.2)
Unknown	74 (63.8)	26 (52)	48 (72.7)
**Triage color tag, *n* (%)**			
White	5 (4.3)	0 (0)	5 (7.6)
Green	58 (50)	13 (26)	45 (68.2)
Yellow	44 (37.9)	28 (56)	16 (24.2)
Red	9 (7.8)	9 (18)	0 (0)

**Table 3 children-09-01794-t003:** Symptoms related to anaphylaxis in 116 pediatric patients referred to the PED. AP = acute patients, NAP = non-acute patients. * Objective symptoms or upper airway compromise: swelling of tongue, swelling/tightness in the throat, difficulty swallowing, hoarse voice, stridor. ** Tachycardia and hypotension defined according to age [12]. In the NAP group, cardiovascular and respiratory symptoms had mostly disappeared at the arrival to the PED and were reported by the patient and/or caregiver.

	Total	AP Group	NAP Group
	*n* = 116	*n* = 50	*n* = 66
**Cutaneous, *n* (%)**	110 (94.8)	48 (96)	62 (93.9)
Urticaria	43 (39.1)	20 (41.7)	23 (37.1)
Angioedema	38 (34.5)	12 (25)	26 (41.9)
Urticaria+Angioedema	27 (24.6)	16 (33.3)	11 (17.7)
Pruritus and/or flushing	2 (1.8)	0 (0)	2 (3.2)
**Upper airway, *n* (%)**	51 (44)	26 (52)	25 (37.9)
Tightness in the throat	19 (37.2)	4 (15.4)	15 (60)
Objective symptoms *	21 (41.2)	18 (69.2)	3 (12)
Nasal obstruction/rhinorrhea	11 (21.6)	4 (15.4)	7 (38)
**Lower airway, *n* (%)**	74 (63.8)	35 (70)	39 (59.1)
Difficulty breathing	27 (36.5)	4 (11.4)	23 (59)
Cough	8 (10.8)	3 (8.6)	5 (12.8)
Wheeze, bronchospasm	30 (40.5)	22 (62.9)	8 (20.5)
Dyspnea	9 (12.2)	6 (17.1)	3 (7.7)
**Cardiovascular, *n* (%)**	7 (6)	5 (10)	2 (3)
Fatigue/dizziness	4 (57.1)	2 (40)	2 (100)
Pale/tachycardic **	2 (28.6)	2 (40)	0 (0)
Hypotension **	1 (14.3)	1 (20)	0 (0)
**Gastrointestinal, *n* (%)**	55 (47.4)	19 (38)	36 (54.5)
Nausea	2 (3.6)	0 (0)	2 (5.6)
Persistent vomiting	43 (78.2)	13 (68.4)	30 (83.3)
Abdominal pain	9 (16.4)	5 (26.3)	4 (11.1)
Diarrhea	1 (1.8)	1 (5.3)	0 (0)

**Table 4 children-09-01794-t004:** PED management of 116 pediatric patients referred for suspected anaphylaxis. AP = acute patients, NAP = non-acute patients; antiH1 = antihistamines; CS = corticosteroids; EAI = epinephrine auto-injector. * The percentages concerning allergological workup and therapy at discharge were calculated after excluding those patients who were admitted. ** The percentages of patients who received a prescription for EAI at PED discharge were calculated after excluding those who already had such prescription and those who were admitted.

	Total	AP Group	NAP Group
	*n* = 116	*n* = 50	*n* = 66
**Diagnosis, *n* (%)**			
Allergic reaction	77 (66.4)	18 (36)	59 (89.4)
Anaphylactic reaction	39 (33.6)	32 (64)	7 (10.6)
**Treatment, *n* (%)**			
Epinephrine IM in PED	14 (12.1)	14 (28)	0 (0)
CS in PED	71 (61.2)	42 (84)	29 (43.9)
AntiH_1_ in PED	73 (62.9)	42 (84)	31 (47)
Only CS and/or antiH_1_ in PED	74 (63.8)	34 (68)	40 (60.6)
No treatment in PED	28 (24.1)	2 (4)	26 (39.4)
Epinephrine IM before PED	0 (0)	0 (0)	0 (0)
CS and/or antiH_1_ before PED	52 (44.8)	19 (38)	33 (50)
**Hospitalization, *n* (%)**			
Discharge	72 (62.1)	13 (26)	59 (89.4)
Short-stay observation	41 (35.3)	35 (70)	6 (9.1)
Admitted	3 (2.6)	2 (4)	1 (1.5)
**Allergological work-up, *n* (%) ***			
no	49 (43.3)	8 (16.7)	41 (63)
Planned/performed	47 (41.6)	36 (75)	11 (16.9)
Recommended	17 (15)	4 (8.3)	13 (20)
**Therapy at discharge ***			
AntiH_1_	98 (86.7)	44 (91.7)	54 (83.1)
CS	60 (53.1)	29 (60.4)	31 (47.7)
EAI available	11 (9.5)	9 (18)	2 (3)
EAI prescribed **	11 (10.8)	10 (25.6)	1 (1.6)

**Table 5 children-09-01794-t005:** Binary logistic regression with IM epinephrine administration as dependent variable and items list in columns 1 as independent variables. Coefficient points to direction of the correlation.

Independent Variables	Coefficient	Significance
Food allergen	−3.661	0.072
Drug allergen	−2.035	0.114
Insect bite allergen	−0.436	0.544
Previous anaphylactic reactions	−0.566	0.352
Symptoms in a recovery phase at arrival	−22.056	0.995
Cutaneous symptoms (ongoing)	1.457	0.751
Respiratory symptoms (ongoing)	2.300	0.027
Cardiovascular symptoms (ongoing)	4.045	0.021
Persistent gastrointestinal symptoms (ongoing)	3.152	0.009

**Table 6 children-09-01794-t006:** Binary logistic regression with diagnosis of anaphylaxis as dependent variable and items list in columns 1 as independent variables. Coefficient points to direction of the correlation.

Independent Variables	Coefficient	Significance
Food allergen	−0.716	0.499
Drug allergen	−0.129	0.833
Insect bite allergen	0.262	0.588
Nut allergy	−0.199	0.492
Cutaneous symptoms (ongoing)	−0.104	0.925
Respiratory symptoms (ongoing)	0.837	0.090
Cardiovascular symptoms (ongoing)	0.278	0.783
Persistent gastrointestinal symptoms (ongoing)	0.372	0.479
Priority Code	1.064	0.009
Symptoms in a recovery phase at arrival	−1.508	0.003

## Data Availability

All clinical data and material are available in our Pediatric Unit.

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
