# Peer review of "Triage Grading and Correct Diagnosis Are Critical for the Emergency Treatment of Anaphylaxis"

_children, 2022, doi:10.3390/children9121794_

Round 1

Reviewer 1 Report

Dear Editor

many thanks for asking me to review this originala article aimed to assess which factors play a major role for 16 a correct diagnosis and an appropriate therapy of anaphylaxis. 

This is a well-written paper and worthy of publication.

However, I strongly invite the authors to update the references and also to refer to more recent papers such as:

1. What is new in anaphylaxis? Acta Biomed. 2020 Sep 15;91(11-S):e2020005. doi: 10.23750/abm.v91i11-S

2. The Timely Administration of Epinephrine and Related Factors in Children with Anaphylaxis. J Clin Med. 2022 Sep 20;11(19):5494. doi: 10.3390/jcm11195494

Author Response

Dear Editor

many thanks for asking me to review this originala article aimed to assess which factors play a major role for 16 a correct diagnosis and an appropriate therapy of anaphylaxis. 

This is a well-written paper and worthy of publication.

Authors’ reply: thank you for appreciating our work.

However, I strongly invite the authors to update the references and also to refer to more recent papers such as:

  1. What is new in anaphylaxis? Acta Biomed. 2020 Sep 15;91(11-S):e2020005. doi: 10.23750/abm.v91i11-S
  2. The Timely Administration of Epinephrine and Related Factors in Children with Anaphylaxis. J Clin Med. 2022 Sep 20;11(19):5494. doi: 10.3390/jcm11195494

Authors’ reply: thank you for suggesting these papers, which were added to the introduction. We also have added more references.

Reviewer 2 Report

The manuscript depicts very good the problem. It reflects the actual situation in most clinical settings. I think that it would be good if there are some recommendations for improving the triage criteria for allergic reactions, e.g. standardized questionnaire in order to increase the rate of recognized allergic cases. The fact, that none of the patients with already prescribed IM epinephrine have received a dose is also important and needs further discussion.

Author Response

The manuscript depicts very good the problem. It reflects the actual situation in most clinical settings. I think that it would be good if there are some recommendations for improving the triage criteria for allergic reactions, e.g. standardized questionnaire in order to increase the rate of recognized allergic cases. The fact, that none of the patients with already prescribed IM epinephrine have received a dose is also important and needs further discussion.

Authors’ reply:

Thank you very much for appreciating our work.

  • Concerning triage interventions, we expanded the discussion about this point and added some concepts to the conclusions
  • Concerning no use of epinephrine auto-injectors before ED referral: discussion of this point was added (end of the discussion section, before study limitations)
